# Phytochemical Constituents, Antimicrobial Properties and Bioactivity of Marine Red Seaweed (*Kappaphycus alvarezii*) and Seagrass (*Cymodocea serrulata*)

**DOI:** 10.3390/foods12142811

**Published:** 2023-07-24

**Authors:** Deep Das, Abimannan Arulkumar, Sadayan Paramasivam, Aroa Lopez-Santamarina, Alicia del Carmen Mondragon, Jose Manuel Miranda Lopez

**Affiliations:** 1Department of Oceanography and Coastal Area Studies, School of Marine Sciences, Alagappa University, Karaikudi 630 003, Tamil Nadu, India; das415269@gmail.com; 2Department of Biotechnology, Achariya Arts and Science College (Affiliated to Pondicherry University), Villiabur, Puducherry 605 110, Tamil Nadu, India; aruul3@gmail.com; 3Laboratorio de Higiene Inspección y Control de Alimentos, Departamento de Química Analítica, Nutrición y Bromatología, Universidad de Santiago de Compostela, 27002 Lugo, Spain; aroa.lopez.santamarina@usc.es (A.L.-S.); alicia.mondragon@usc.es (A.d.C.M.); josemanuel.miranda@usc.es (J.M.M.L.)

**Keywords:** phytochemicals, seaweed, seagrass, *Kappaphycus*, *Cymodocea*, radical scavenging, phytochemical analysis, antioxidant activity, antibacterial properties, nutritional value

## Abstract

The present work was performed to evaluate the levels of phytochemical constituents and the antioxidant and antibacterial properties of marine red seaweed (*Kappaphycus alvarezii*) and seagrass (*Cymodocea serrulata*). Quantitative phytochemical analysis, antioxidant activity and antimicrobial activity against five potential pathogenic bacteria was investigated. In each case, we found the presence of flavonoids, tannins, phenolic compounds, glycosides, steroids, carbohydrates and ashes. Alkaloids were only found in *K. alvarezii*, though they were not found in *C. serrulata*. The antimicrobial properties of both *K. alvarezii* and *C. serrulata* chloroform extracts were found to be antagonistically effective against the Gram-positive bacteria *Bacillus subtilis* and the Gram-negative bacteria *Vibrio parahaemolyticus*, *Vibrio alginolyticus, Vibrio harveyi* and *Klebsiella pneumoniae*. GC-MS analysis revealed the presence of 94 bioactive compounds in *K. alvarezii* and 104 bioactive compounds in *C. serrulata*, including phenol, decane, dodecane, hexadecane, vanillin, heptadecane, diphenylamine, benzophenone, octadecanoic acid, dotriaconate, benzene, phytol, butanoic acid and 2-hydroxyl-ethyl ether, which all played important roles in antioxidant and antibacterial activities. Thus, in view of the results, both *K. alvarezii* and *C. serrulata* could be considered to be sources of ingredients with appreciable nutritional and medicinal value.

## 1. Introduction

Marine organisms are valuable sources of bioactive compounds used by both the food and pharmaceutical industries. Bioactive compounds can be obtained from a wide range of marine foods. Nowadays, more than 36,000 compounds with potential effects on human health have been isolated from marine organisms [1]. Significantly, such bioactive compounds can minimize chronic non-communicable disease risk by reducing the onset of inflammation and oxidation [2]. In recent years, seaweeds have been reported to be an important source of bioactive compounds [3,4]. Another marine organism that contains a large variety of bioactive compound is seagrass, which, although it is less used than seaweeds, has been used in food and medicine by populations of coastal region [5].

Seaweeds are a group of autotrophic, halophytic and complex communities that live in marine environments and have the potential to be used as renewable resources [6,7]. Biologically, they are classified as either Phaeophyta (brown algae), Rhodophyta (red algae) or Chlorophyta (green algae) [4]. Seaweeds grow in salt water, especially in shallow coastal waters, and can be obtained for human consumption in both wild and cultivated forms [8]. Although, in recent decades, the use of seaweeds as food has increased in other parts of the world [4], seaweeds are still mostly consumed in Asian countries, such as Japan, China or South Korea [4]. Another marine source of bioactive compounds with a broad spectrum of beneficial activities for human health is seagrass [9]. Seagrasses are submerged flowering marine angiosperms that live their full lifecycles while submerged in marine environments, and they are the primary producers. They are the only flowering plants to recolonize the sea band, are highly productive and play an important ecological role in marine environments [5]. Seagrasses are found in all coastal areas around the world, except in Antarctica [10,11].

Seaweed consumption has numerous advantages for human health due to its dietary fiber, protein, essential fatty acid, vitamin and essential mineral content [12]. Seaweed’s proximate and nutritional composition varies and is affected by a large variety of factors, including the seaweed species, the geographic area of origin, solar intensity or the seawater temperature [4]. Besides their uses as foods, other uses of seaweeds have been widely increased in recent decades. Indeed, nowadays, seaweeds are also used as fertilizers and cosmetics, and their extracts are used in pharmaceutical industries as a fresh source of bioactive compounds with a wide range of medicinal properties [13]. Regarding seagrass, they are used in some countries as a source of food for human consumption, and their macronutrient compositions are similar to those of plants of terrestrial origin [5].

Regarding their bioactive compounds contents, seaweeds are one of the richest sources of bioactive primary and secondary metabolites, which are characterized by beneficial biological activities [6]. Besides advantages for human health, seaweeds are known for their potential natural antioxidant, antiviral, antiobesity, antitumor and antimicrobial properties [12,13,14]. With respect to seagrasses, they have been used in folk medicine to treat infections caused by pathogenic microbes, fever, inflammation, muscle pain, skin disease, viruses, diarrhea, diabetes, wound healing, sedation and cancer [14], as well as to design tranquillizers for babies or remedies to treat ray stings [5]. Seagrasses are rich sources of secondary metabolites, such as alkaloids, flavonoids, terpenoids, tannins, steroids, and, in particular, phenolic compounds [15], which contribute to pigmentation, growth, reproduction and pathogen resistance, and they also act as defensive mechanisms against other aquatic threats [16].

Among seaweeds, *Kappaphyus* spp. is a commercially important red seaweed that is cultivated in tropical countries, such as the Philippines, Indonesia and Malaysia, as well as in many countries in Eastern Africa [13], because it is relatively easy to cultivate and has short production cycles and low production costs [13,17]. It is also a common food source for local people and is believed to have various beneficial effects. In India, the southeastern coast has a unique marine habitat that supports a great variety of seaweed species within the intertidal zone and shallow- and deep-water regions of the ocean. Specifically, *Kappaphycus alvarezii* (commercially known as “cottoni”) achieves good growth along the shores of the Kanyakumari and Ramanthapuram Districts of Tamil Nadu, India [18]. *K. alvarezii* has a high economic value, since it is the principal source of the commercial hydrocolloid κ-carrageenan and contains various inorganic and organic compounds that are beneficial for human health [13,19]. k-carrageenan is used in pharmaceuticals, cosmetics, textiles and organic fertilizers, as well as in the food industry [19].

Among the different genii that form seagrasses, *Cymodocea*, which is part of the Potamogetonaceae family, is represented globally by four species: *Cymodocea rotundata*, *Cymodocea serrulata*, *Cymodocea angustata* and *Cymodocea nodosa* [20]. *C. serrulata* is commonly found in the coastal areas of the tropical Indo–West Pacific region. *C. serrulata* can be differentiated from other seagrass species based on their shoots, which have distinctive open leaf scars, as well as triangular and flat leaf sheath fibrous roots on the shoot and serrated leaf tips [10].

Although the composition of *K. alvarezii* was previously reported as being of the same origin, while the phytochemical composition of *C. serrulata* was also previously reported, it should be considered that the compositions of properties of both seaweed and seagrasses widely vary depending on a large variety of factors [4,5]. Thus, in a scenario in which important factors, such as ocean acidity, salinity or temperature, are changing [21], it is important to have updated data regarding seaweed and seagrasses composition and bioactivities. Hence, the aim of the present study was to evaluate the phytochemical constituents, antioxidant activity and antibacterial activity present in seaweeds and seagrass. The chemical compounds present in both *K. alvarezii* and *C. serrulata* were also determined.

## 2. Materials and Methods

### 2.1. Collection, Identification and Processing

The red seaweed *K. alvarezii* and the seagrass *C. serrulata* were obtained from Thondi coastal waters (Latitude: 9°44″ N and Longitude: 79°00″ E) in Palk Bay, India. Freshly collected seaweed (*K. alvarezii*) and seagrass (*C. serrulata*) (Figure 1) were cleaned thoroughly in seawater and transported to the laboratory in a portable cooler that was protected from sunlight within 1 h of collection. The epiphytes, necrotic parts, muds, dust and other debris were removed via thorough washing with fresh water and double-distilled water. Next, they were shade-dried in an oven (Memmert, Roentgen, Germany) at 25 ± 2 °C for one week, ground into fine powder and stored at room temperature in an airtight container (Tarsons, Chennai, India) until analysis. The collected seaweed and seagrass were identified according to those standards established in the standard manual of Rao [22].

### 2.2. Preparation of Extracts

The seaweed and seagrass extracts were made by adding 5 g of dried seaweed or seagrass powder into 50 mL of three different solvents—chloroform, ethanol and distilled water—in a conical flask and placing the solution in a dark bottle in light agitation (AGIMATIC-N, J.P. SELECTA S. A, Barcelona, Spain) for 7 days. After that, the extracts were filtered through Whatman No. 1 filter papers and sterile cotton wools, and the supernatants were stored at 4 °C for future use [23,24].

### 2.3. Determination of Alkaloids

The alkaloid contents of *K. alvarezii* and *C. serrulata* were determined via the method proposed by Hikino et al. [25]. Next, 1 mL of test extract phosphate buffer (5 mL, pH 4.7) was added to 5 mL of bromocresol green solution, and the mixture was shaken vigorously added with 4 mL of chloroform. The extracts were collected in a 10-milliliter flask. The absorbance of the complex in chloroform was measured at 470 nm using a UV-Vis spectrophotometer (Shimadzu, Kyoto, Japan) against a prepared blank, as described above but without extract. Atropine (Sigma-Aldrich, St. Louis, MO, USA) was used as the standard, and the assay was compared to atropine equivalents.

### 2.4. Determination of Flavonoids

Total flavonoid content was determined via the aluminum chloride method [26] and using catechin (Sigma-Aldrich) as standard. Next, 1 mL of test sample and 4 mL of water were added to a volumetric flask (10-milliliter volume). After 5 min, 0.3 mL of 5% sodium nitrite and 0.3 mL of 10% aluminum chloride (Sisco Research Laboratories, Mumbai, India) were added. After 6 min of incubation at room temperature, 2 mL of 1-molarity sodium hydroxide (Sisco Research Laboratories) was added to the mixture. Afterwards, the final volume was brought to 10 mL via addition of distilled water. The absorbance of the reaction mixture was spectrophotometrically measured at 510 nm against a blank using a UV-Vis spectrophotometer (Shimadzu). The results obtained were expressed as catechin equivalents (mg catechin/g dried extract).

### 2.5. Determination of Tannins

The total tannin content extracts were determined according to the Julkunen–Titto [27] method. Firstly, 50-microliter extracts were mixed with 1.5 mL of 40% vanillin (Sisco Research Laboratories) (prepared with methanol), and 750 µL of HCl was then added. The solution was shaken vigorously and left at room temperature for 20 min in darkness. The absorbance of the mixtures was measured at 500 nm using a spectrophotometer (Shimadzu). A calibration curve was constructed using catechin (Sigma-Aldrich) in the range of 20–200 mg/L.

### 2.6. Determination of Phenolic Compounds

The total phenolic content in solvent extracts was determined using Folin–Ciocalteu’s reagent, as proposed by Sangeeta and Vrunda [28]. During the procedure, different concentrations of the extracts were mixed with 0.4 mL of Folin–Ciocalteu’s reagent (Sigma-Aldrich) (diluted 1:10 *v*/*v*). After 5 min, 4 mL of a sodium carbonate solution was added. The final volume of the tubes was brought to 10 mL by adding distilled water and left for 90 min at room temperature. The absorbance of the samples was measured against a blank sample at 750 nm using a spectrophotometer (Shimadzu). A calibration curve was constructed using 1,2-dihydroxybenzen (catechol) (Sigma-Aldrich) solutions as standards, and the total phenolic content of the extract was expressed in terms of the mg of catechol per g of dry weight.

### 2.7. Determination of Cardiac Glycosides

The cardiac glycoside content was determined using Buljet’s reagent based on the method reported by El-Olemy et al. [29]. Firstly, 1 g of the fine powder of *K. alvarezii* and *C. serrulata* was soaked in 10 mL of 70% MeOH for 2 h and filtered. The extract obtained was then purified using lead acetate and Na_2_HPO_4_ solution before the addition of freshly prepared Buljet’s reagent (containing 95 mL of aqueous Picric acid and 5 mL of 10% aqueous NaOH) (Sigma-Aldrich). The difference between the intensity of colors of the experiment and blank samples gives an absorbance of 217 nm using a spectrophotometer (Shimadzu), which was used to calculate the concentration of glycosides.

### 2.8. Determination of Steroids

The steroid content was determined by Ejikeme et al. [26]. Firstly, 1 mL of test extract of the steroid solution was transferred into 10-milliliter volumetric flasks. Sulfuric acid (Sisco Research Laboratories) (4 N, 2 mL) and iron (III) chloride (Sisco Research Laboratories) (0.5% *w*/*v*, 2 mL) were added, followed by potassium hexacyanoferrate (III) solution) (Sisco Research Laboratories) (0.5% *w*/*v*, 0.5 mL). The mixture was heated via a Memmert WTB water bath (Memmert, Schutzart, Germany) that was maintained at 70 ± 20 °C for 30 min, along with shaking, and afterwards diluted to the mark with distilled water. The absorbance was measured using a 780 nm spectrophotometer (Shimadzu) against the reagent blank.

### 2.9. Determination of Carbohydrates

Carbohydrate content was estimated based on the phenol-sulfuric acid method [30]. In brief, 200 mg of a powdered sample, which was weighed using an analytical balance (OHAUS GA200, Nänikon Switzerland), was hydrolyzed by adding 5 mL of 2.5 N HCl. The sample was kept in boiling water, and after 3 h, the solution was neutralized with solid Na_2_CO_3_ until effervescence ceased. The solution was made of up to 50 mL and centrifuged at 8000 rpm for 10 min in a centrifuge (Remi Lab World, Mumbai, India). Afterwards, the supernatant was aliquoted and brought up to 1 mL using deionized water, to which component 1 mL of phenol and 5 mL of 96% sulfuric acid (Sisco Research Laboratories) were previously added. After mixing the solution, it was kept in a water bath at 25 ± 1°C for 20 min. The absorbance was measured at 490 nm using a UV-Vis spectrophotometer (Shimadzu) against the reagent blank.

### 2.10. Ash Content

The ash content was determined using the method of Yemm and Willis [30]. Firstly, 2 g of each sample was taken and weighed accurately using a Cobos CB balance (Barcelona, Spain) in a clean silica dish. The dish was first heated over a low burner flame. Next the dish was transferred to a SNOL muffle furnace (Utena, Lithuania) maintained at 500–550 °C for 3–5 h. The ash residue obtained was then cooled in a desiccator and weighed on a balance. The percentage of total ash content was calculated via the following formula:Total Ash Percent of plant sample (%) = [Weight of dry ash residue (g) ÷ Weight of plant sample (g)] × 100

### 2.11. Hydrogen Peroxide Radical Scavenging Activity

The antioxidant activity of seaweed and seagrass extracts was evaluated based on the hydrogen peroxide radical scavenging activity, as described by Ebrahimzadeh et al. [31]. The seaweed and seagrass extracts (100 µg/mL) were reacted with 0.6 mL of 40 mM H_2_O_2_ solution prepared in phosphate buffer (pH 7.4) (Sisco Research Laboratories). After incubation at 37 °C for 10 min, absorbance was measured at 230 nm using a UV-Vis spectrophotometer (Shimadzu). Phosphate buffer was used as the corresponding blank solution. A similar procedure was repeated using distilled water instead of the extract, which served as a control. Ascorbic acid (Sigma-Aldrich) (20–100 µg/mL) was used as a standard.

### 2.12. In Vitro Antibacterial Activity of Seaweed and Seagrass against Human Pathogenic Bacteria

The antibacterial activity of seaweed and seagrass extracts was evaluated via the well diffusion method using a Muller–Hinton agar (Hi-Media, Mumbai, India). Approximately 100 µL of 10^5^ CFU/mL of diluted inoculum of bacterial culture was applied to the surface of Muller–Hinton agar plates. The Muller–Hinton agar well was made using a well borer under aseptic conditions and filled with *K. alvarezii* and *C. serrulata* extracts, and methanol served as a positive control. The plates were incubated at 37 °C for bacterial growth, and the antibacterial activity of the seaweed and seagrass samples was evaluated by measuring the zone of inhibition (mm) in relation to the tested pathogenic bacteria. All experiments were performed in triplicate, and the data are expressed as the mean values of the experiments.

### 2.13. Characterization of the Active Compound by Gas Chromatography-Mass Spectrometry (GC-MS)

The crude extracts of *K. alvarezii* and *C. serrulata* were loaded into a silica gel (Hi-Media) packed column (20 cm length and 2 cm diameter) and eluted using n-hexane: ethyl acetate (50:50 *v*/*v*) (Sigma-Aldrich). The fractions were analyzed via a gas chromatograph GC-2010 interfaced with a quadrupole mass spectrometer QP-2010 (Shimadzu, Japan) analyzer, which used an Rtx-PCB capillary column (60 m × 0.25 mm i.d., 0.25 mm film thickness, Resteck, Bellefonte, PA, USA). Helium with a purity of 99.99% was used as the carrier gas at a flow rate of 1 mL/min. Next, 1 mL of extract was injected in spilt mode using an autosampler (Shimadzu). The injector port, interface and ion source temperature were set at 250, 270 and 230 °C, respectively. The mass spectrometer was operated in electron ionization (EI) mode at 70 eV and at an emission current of 60 mA. Full scan data were obtained in a mass range of 50–500 *m*/*z*. Interpretation of mass spectrum data was performed using the National Institute Standard and Technology (NIST) database.

### 2.14. Statistical Analysis

All determinations were given in terms of the mean ± standard deviation (SD). The results obtained were compared via one-way analysis of variance (ANOVA). The significance of the difference between means was determined via Duncan’s multiple range test (*p* < 0.05) using SPPS version 14 (Chicago, IL, USA).

## 3. Results and Discussion

### 3.1. Phytochemical Analysis

Phytochemical analysis of *K. alvarezii* and *C. serrulata* revealed the presence of alkaloids (only in the case of *K. alverazii*), flavonoids, tannins, phenolic compounds, glycosides, steroids, carbohydrates and ashes. Among the six phytochemicals present in *K. alvarezii,* higher contents were found for phenolic compounds (3.39 ± 0.41 mg/g) and tannins (2.94 ± 0.41 mg/g). Both phenolic compounds and tannins have important roles as bioactive compounds. Phenols have important antioxidant, antimicrobial, anti-inflammatory and anticancer activities [32,33], whereas tannins are reported to have antiviral, antibacterial and antioxidant activities [34,35]. Among the five phytochemicals present in *C. serrulata*, the highest contents were found for glycosides (2.47 ± 0.41 mg/g) and flavonoids (2.11 ± 1.40 mg/g) (Table 1). With respect to glycosides, it was reported that they have antioxidants and anti-inflammatory activities, which find application in the prevention and managements of several human diseases [34]. Flavonoids also make up an important phytochemical group due to their antimicrobial, antiviral, antioxidant and spasmolytic activities [35]. These constituents significantly contribute to the biological activity of seaweeds and seagrass [36]. Similar observations were also made by other works [37,38], which found tannins, flavonoids, phenolic compounds, carotenoids and polysaccharides in both seaweed and seagrasses.

In the present study, *K. alvarezii* showed a higher tannin content (2.94 ± 0.41 mg catechin equivalent (CAE)/g) than *C. serrulata* (1.94± 0.85 mg CAE/g). Similarly, Deyad and Ward [35] reported similar tannin content in the brown seaweed *Dictyota dichotoma* (2.12 ± 0.45 mg CAE/g), whereas Domettila et al. [32] reported a higher presence of tannins in the red seaweed *Sargassum wightii* (27.54 ± 0.54 mg CAE/g). In previous studies [9], the presence of tannins in *C. serrulata* (264.71 mg/mL tannic acid equivalence) was reported. Similarly, another work reported the presence of tannins in the seagrass *Syringodium isoetifolium* (80.65 ± 5.64 mg CAE/g [39]. Tannins are polyphenols, which have a large influence on the nutritive value of humans and animals due to their antimicrobial, anti-inflammatory, and astringent activities [9].

Flavonoid content was similar in both *K. alvarezii* and *C. serrulata*, although in global terms, it was found in lower amounts than in previous works. Vaghela et al. [40] found a much higher flavonoid content (15.26 ± 0.95 mg CAE 100 g^−1^). Similarly, Smadi et al. [41] reported the flavonoid content of *C. nodosa* to be 3.98 ± 0.03 mg CAE/g, which is comparatively higher than the results of the present study.

*K. alvarezii* had an alkaloid content of 1.91 ± 0.58 mg CAE/g. Similarly, Domettila et al. [32] showed an alkaloid content of 1.32 ± 0.02 mg CAE/g in *Ulva reticulata*. Previously, Alghazeer et al. [42] reported the alkaloid content in the brown algae species *Cystoseira compressa* and *Sargassum hornschuchii* to be 4125.00 ± 180.28 mg/g DW and 3708.33 ± 152.75 mg/g DW, respectively. Alkaloids have been proven to have antiplasmodic, antimicrobial, and cytotoxic properties [42].

The phenolic compound content in seaweeds is, in part, responsible for their scavenging activity, which protects them against lipid oxidation [43]. In this work, *K. alvarezii* showed a higher phenolic content (3.39 ± 0.45 mg gallic acid equivalents (GAE)/g) than *C. serrulata* (1.01 ± 0.39 mg GAE/g). Previously, other authors reported a significantly higher content of phenolic compounds in both *K. alvarezii* (3.14 ± 0.14 mg GAE/g) [44] and *Kappaphycus striatum* (7.24 ± 0.21 mg GAE/g) [45]. Regarding *C. serrulata*, the results obtained in the current work were also significantly lower than those reported by Libin et al. [17] for *C. serrulata* (2.98 ± 0.12 mg GAE/g) and *Cynodocea rotundata* (2.04 ± 0.1) [46]. The phenolic contents of seaweed and seagrass depend on the solvent used to analyze the extraction process, environment, habitat and biomass.

The presence of steroids in seaweed *K. alvarezii* (2.51 ± 0.15 mg/g) was higher than that recorded in seagrass *C. serrulata* (1.60 ± 0.24 mg/g). Previously, another study showed that the presence of steroids in seaweed *C. elongata* was 2.27 ± 0.26 mg/g [47]. Kumar et al. [48] also reported the presence of steroids in *Champai parvula* (24.30 ± 0.11 mg/g). Previously, the presence of steroids in *Himanthalia elongata* (2.64 ± 2.21 mg/g) was reported [49]. In previous studies, Kannan et al. [50] reported the presence of steroids in *C. rotundata* (2.37 ± 1.27 mg/g). Similarly, Tango et al. [51] also reported the presence of steroids in the seagrass *Haludole pinifolia* (5.62 ± 0.76 mg/g). Steroids isolated from seaweed and seagrass have medicinal values, such as antihelmintic, antioxidant, antimicrobial and antiviral activities [52].

*K. alvarezii* showed a glycoside content of 1.88 ± 0.11 mg/g, while in *C. serrulate*, glycoside content was reported to be 2.47 ± 0.28 mg/g. Previously, Kumar et al. [48] reported the presence of glycosides in seaweed *Cymodocea parvula* (35.33 ± 0.14 mg/g). Similarly, Prabakaran et al. [52] reported the presence of glycosides in *Chorella vulgaris* (5.75 ± 0.23 mg/g). Deyad and Ward [35] also reported the presence of glycosides in the seaweed *D. dichotoma* (2.14 ± 0.15 mg/g). A previous work performed by Regalado et al. [53] reported the presence of glycosides in *Thalassia testudinum* (4.61 ± 1.60 mg/g). Glycosides are well known for being able to lower blood pressure in humans [48].

With respect to carbohydrates and ash content, the carbohydrate content of *K. alvarezii* was 2.57 ± 1.89 mg/g DW, while that of *C. serrulata* was 1.44 ± 1.75 mg/g DW. The wide variation in the carbohydrate content observed between seaweed and seagrass might be due to the influence of different factors, such as salinity, temperature and sunlight intensity. Regarding ash, *K. alvarezii* had a higher ash content (8.5 ± 0.95 g/100 g) than *C. serrulata* (6.9 ± 0.49 g/100 g). High ash content showed the presence of appreciable amounts of diverse minerals found in both seaweed and seagrass.

### 3.2. Antioxidant Activity

Antioxidant effectiveness is measured by monitoring the inhibition of oxidation of a suitable substrate [15]. In biological systems, antioxidant effectiveness is classified into two groups: evaluation of lipid peroxidation and measurement of free radical scavenging ability [31]. The in vitro antioxidant activiies of *K. alvarezii* and *C. serrulata* extracts were evaluated based on hydrogen peroxide radical scavenging activity, and *K. alvarezii* had higher scavenging activity (27.9 ± 0.1%) than *C. serrulata* (22.1 ± 0.1%). Regarding *K. alvarezii*, the results obtained were higher than those previously reported by other authors, such as Farah et al. [37] or Chew et al. [54], who reported lower (18.34 ± 0.57% and 11.8 ± 5.7%, respectively) 2,2-Diphenyl-1-picrylhydrazyl (DPPH) scavenging activity. Regarding *C. serrulata*, the DPPH scavenging activity results obtained were lower than those obtained by Kannan et al. [50] (61.85 ± 0.95%) regarding free radical scavenging activity from the same seagrass species, though higher than those results reported by Rengasamy et al., [33] (6.65 ± 0.12%) for other *Cymodocea* species, such as *C. rotundata.*

### 3.3. Antimicrobial Activity

The antibacterial activity of both *K. alvarezii* and *C. serrulata* were investigated using chloroform extracts based on those reported by Pusparaj et al. [14], who reported that the best inhibitory effects of *K. alvarezii* were reported using chloroform extracts. The antibacterial activities of both *K. alvarezii* and *C. serrulata* depend on the presence of bioactive compounds, phenolic content and free radical scavenging activity [55]. In all cases, inhibitory activities against the five pathogenic bacteria investigated were detected (Table 2). The higher inhibitory activity was observed in *K. alvarezii* (26 ± 0.03 mm) against *Bacillus subtilis,* as well as in the case of *C. serrulate*, which exhibited maximum inhibitory activity (26 ± 0.08 mm) against *Vibrio parahaemolyticus*. The chloroform extract of *K. alvarezii* showed maximum activity of 26 ± 0.03 mm against *B. subtilis* at 100 µg/mL, and *C. serrulata* showed maximum activity of 26 ± 0.08 mm against *V. parahaemolyticus* at 100 µg/mL and minimum activity of 22 ± 0.01 mm and 20 ± 0.04 mm against *Vibrio alginolyticus* at 100 µg/mL in both *K. alvarezii* and *C. serrulata*, respectively (Table 2).

Jaswir et al. [55] reported maximum inhibitory activity (12 ± 1.02 mm) against *B. subtilis* using the methanolic extract of *K. alvarezii*. Similarly, Pusparaj et al. [14] reported the antibacterial activity of *K. alvarezii* against six human pathogens: *Staphylococcus aureus*, *B. subtilis*, *Lactobacillus acidophilus*, *Pseudomonas aeruginosa*, *Escherichia coli* and *Proteus mirabillis*. He also reported that the best activity was recorded in chloroform extracts. Kumar et al. [56] studied the antibacterial activity of *C. serrulata* against four fish-borne pathogens, namely *Bacillus cereus*, *B. subtilis*, *E. coli* and *Micrococcus luteus*, and reported that *C. serrulata* was effective against several *Bacillus* species.

### 3.4. Presence of Bioactive Compounds

The GC-MS running time for the n-hexane:ethyl acetate (50:50 *v/v*) extracts of *K. alvarezii* and *C. serrulata* was 30 min. The target mass ions (*m*/*z*) and retention times (min) of all identified compounds in *K. alvarezii* and *C. serrulata* are shown in Table 3 and Table 4. The results show that *K. alvarezii* extracts contained 94 different bioactive compounds, including phenol, decane, dodecane, hexadecane, vanillin, heptadecane, diphenylamine, benzophenone, octadecanoic acid, dotriacontane and benzene (Table 3). On the other hand, *C. serrulata* was found to contain 104 different bioactive compounds, including tetradecane, dodecanal, diphenylamine, heptadecane, phytol, butanoic acid, 2-hydroxy-, ethyl ester, dodecane and benzene (Table 4). These compounds were responsible for the antioxidant and antibacterial activities of both *K. alverazii* and *C. serrulata*.

Datchanamurthy et al. [57] reported that red algae (*Acoathophora deilei*) contain major common components, such as hexadecanoic acid methyl ester, dibutyl phthalate, 2-ethyl butyric acid, octadecyl ester, 9-octadecanoic acid, methyl ester and 1,2-benzendicarboxylic acid. Similarly, Anitha et al. [36] also studied the presence of phenols, hexadecanoic acid, n-hexadecanoic acid, tridecanoic acid, n-nonadecanoic acid and benzene reported to be present in red algae (*Gracilaria cervicornis*). Pushpabharathi et al. [9] reported that nine bioactive components were present in seagrass (*C. serrulata*): hexahydofarnesyl acetone, hexadecanoic acid, methyl ester, n-hexadecanoic acid, tetradecanoic acid, pentadecanoic acid, cholestesta 4,6 dien 3-ol and stigmasterol.

## 4. Conclusions

The red seaweed *K. alvarezii* and seagrass *C. serrulata* examined in the present study were found to possess rich sources of phytochemicals. The antioxidant properties of both seaweed and seagrass reveal that they have appreciable levels of protection against free radicals.

GC-MS analysis revealed the presence of large active metabolites (94 in the case of *K. alvarezii*, and 104 in the case of *C. serrulate*), such as phenol, decane, dodecane, hexadecane, vanillin, heptadecane, diphenylamine, benzophenone, octadecanoic acid, dotriacontane and benzene, in both red seaweed and seagrass. In view of the results obtained, both *K. alvarezii* and *C. serrulata* could be employed as potential marine-sourced drugs and may be used in the pharmaceutical and food processing industries as sources of ingredients with appreciable medicinal value. Since both red seaweed and seagrass were found to be good sources of essential phytochemicals, their commercial value can be enhanced by marketing them to consumers as value-added products.

## Figures and Tables

**Figure 1 foods-12-02811-f001:**
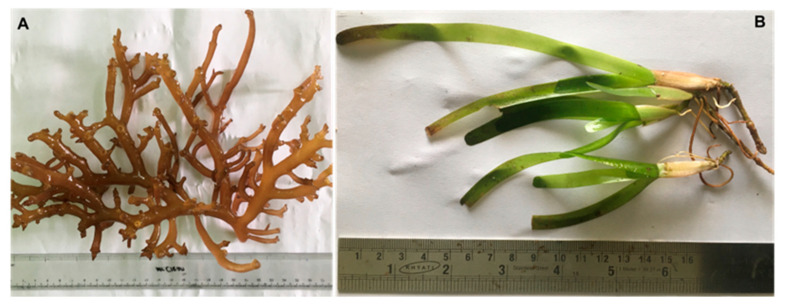
Images of *Kappaphycus alvarezii* (**A**) and *Cymodocea serrulata* (**B**), respectively.

**Table 1 foods-12-02811-t001:** Steroids, tannins, flavonoid, glycosides, alkaloids and phenolic compounds of *Kappaphycus alvarezii* and *Cymodocea serrulata*.

Parameters	*K. alverazii*	*C. serrulata*
Alkaloids (ATE/g dry wt)	1.91 ± 0.58 *	-
Flavonoids (CAE/g dry wt)	1.63 ± 0.73	2.11 ± 1.40
Tannins (CAE/g dry wt)	2.94 ± 0.41 *	1.94 ± 0.85
Phenolic compounds (GAE/g dry wt)	3.39 ± 0.45 *	1.01 ± 0.39
Glycosides (mg/g dry wt)	1.88 ± 0.11	2.47 ± 0.28 *
Steroids (mg/g dry wt)	2.51 ± 0.15 *	1.60 ± 0.24
Carbohydrates (% DW)	2.57 ± 1.89	1.44 ± 1.75
Ash (% DW)	8.5 ± 0.95	6.9 ± 0.49
Antioxidant activity	27.9 ± 0.1	22.1 ± 0.1

Values are means of three analyses of the extracts ± standard deviation (n = 3); CAE: catechin equivalent; GAE: gallic acid equivalents; ATE: atropine equivalent. * Average values are significantly higher than counterparts (*p* < 0.05).

**Table 2 foods-12-02811-t002:** Antibacterial activity of *K. alvarezii* and *C. serrulata* in chloroform extract against human pathogenic bacteria.

Human Pathogenic Bacteria	Concentration (µg/mL)	Seaweed Extract	Seagrass Extract
		Zone of Inhibition (mm)	
*Bacillus subtilis*	100	26 ± 0.03	25 ± 0.16
*Klebsiella pneumoniae*	100	23 ± 0.01	22 ± 0.20
*Vibrio alginolyticus*	100	22 ± 0.01	20 ± 0.04
*Vibrio parahaemolyticus*	100	24 ± 0.02	26 ± 0.08
*Vibrio harveyi*	100	24 ± 0.10	22 ± 0.01

Date were expressed as the mean ± SD values of triplicates (n = 3).

**Table 3 foods-12-02811-t003:** List of compounds identified by analyzing the purified extracts of *K. alvarezii* using GC-MS analysis.

No.	Name	Retention Time (min)	Base *m*/*z*
1	Phenol	4.119	94.05
2	Cyclopropyl methyl carbinol	4.170	58.05
3	Decane	4.347	57.05
4	Butanoic acid, 2-hydroxy-, ethyl ester	4.429	59.05
5	2-Methylpentyl formate	4.457	56.05
6	Benzene, 1,4-dichloro-	4.530	145.95
7	Cyclopentane, 1,2-dimethyl-, cis-	4.568	70.10
8	Dodecane, 2,6,11-trimethyl-	5.158	57.05
9	Undecane, 5-methyl-	5.237	57.10
10	Ethane, hexachloro-	5.395	116.90
11	Dodecane, 2,6,10-trimethyl-	5.803	57.05
12	3-Ethyl-3-methylheptane	5.890	57.05
13	Naphthalene	6.994	128.10
14	Dodecane	7.201	57.05
15	Benzaldehyde, 2,5-dimethyl-	7.450	133.10
16	Octadecanoic acid, phenyl ester	7.517	94.05
17	Benzene, 1,3-bis(1,1-dimethylethyl)-	7.986	175.15
18	Undecane, 2,4-dimethyl-	8.093	57.10
19	Dodecane, 4,6-dimethyl-	8.317	57.05
20	Hexadecane	8.437	57.10
21	Formamide, N-phenyl-	8.884	121.05
22	Dodecane, 2,6,10-trimethyl-	8.942	57.10
23	Chloroxylenol	9.762	121.10
24	Benzene, 1-cyclobuten-1-yl-	9.844	129.10
25	Hexadecane	9.907	57.05
26	Vanillin	9.950	151.05
27	Heptadecane	10.186	57.05
28	Dodecane, 2,6,10-trimethyl-	10.690	57.10
29	1-Dodecanol	10.864	55.05
30	Nonadecane	11.005	85.10
31	Decane, 1-bromo-2-methyl-	11.044	57.05
32	Heneicosane	11.113	71.10
33	Nonadecane	11.159	57.05
34	2,4-Di-tert-butylphenol	11.351	191.15
35	Hexadecane	11.655	57.05
36	Hexadecane	12.350	57.05
37	Diphenylamine	12.664	169.15
38	Benzophenone	12.754	105.05
39	3-Hydroxydiphenylamine	13.216	185.10
40	Hexadecane, 2,6,10,14-tetramethyl-	13.328	57.10
41	Heptadecane	13.476	57.05
42	Dodecane, 2,6,10-trimethyl-	13.540	71.10
43	Heneicosane	13.590	71.10
44	Heneicosane	13.681	57.10
45	Decane, 1-iodo-	13.751	71.10
46	Heptadecane, 8-methyl-	13.940	71.10
47	Heneicosane	14.061	71.10
48	Tetradecanoic acid	14.148	57.05
49	Formamide, N,N-diphenyl-	14.487	168.10
50	Heneicosane	14.549	57.05
51	p-(Benzylideneamino)phenol	14.651	196.10
52	Isopropyl myristate	14.836	60.00
53	Carbamic chloride, diphenyl-	14.990	196.10
54	2-Pentadecanone, 6,10,14-trimethyl-	15.036	57.05
55	Phenoxazine	15.211	183.10
56	1,2-Benzenedicarboxylic acid, bis(2-methylpropyl) ester	15.305	149.05
57	Heneicosane	15.445	57.05
58	Hexadecane	15.569	57.05
59	Tetrapentacontane	15.735	71.10
60	7,9-Di-tert-butyl-1-oxaspiro(4,5)deca-6,9-diene-2,8-dione	15.837	57.05
61	3-Hydroxydiphenylamine	15.925	185.10
62	Benzoic acid, 2-benzoyl-, methyl ester	16.008	163.05
63	n-Hexadecanoic acid	16.227	73.05
64	7,9-Di-tert-butyl-1-oxaspiro(4,5)deca-6,9-diene-2,8-dione	16.414	57.05
65	Heneicosane	16.547	57.05
66	Cyclic octaatomic sulfur	16.953	63.95
67	Palmitic acid, TMS derivative	17.018	117.05
68	Dotriacontane	17.125	57.05
69	Tetrapentacontane	17.472	57.05
70	Pentatriacontane	17.560	85.10
71	Octadecane, 3-ethyl-5-(2-ethylbutyl)-	17.635	71.10
72	Dotriacontane	17.723	57.05
73	Dotriacontane	17.820	71.10
74	Octadecanoic acid	18.063	73.05
75	Tetrapentacontane	18.120	71.10
76	Tetrapentacontane	18.193	71.10
77	Heneicosane	18.374	57.05
78	Tetracosane	19.231	57.10
79	Tetrapentacontane	19.605	71.10
80	1-Heptadecanamine	19.985	85.10
81	Heneicosane	20.056	57.05
82	Benzenemethanamine, N-hydroxy-N-(phenylmethyl)-	20.519	91.05
83	9-Octadecenenitrile, (Z)-	20.833	55.05
84	Bis(2-ethylhexyl) phthalate	21.292	149.05
85	13-Docosenamide, (Z)-	21.461	59.05
86	9-Octadecenamide, (Z)-	21.520	59.05
87	Dotriacontane	21.670	57.05
88	Tetracontane	22.656	57.05
89	13-Docosenamide, (Z)-	23.720	59.05
90	Squalene	24.338	69.05
91	Tetrapentacontane	25.429	57.05
92	13-Docosenamide, (Z)-	26.983	59.00
93	Dotriacontane	27.136	57.05
94	Cholesterol	28.552	386.35

**Table 4 foods-12-02811-t004:** List of compounds identified by analyzing the purified extracts of *C. serrulata* using GC -MS analysis.

No.	Name	R. Time (min)	Base *m*/*z*
1	1-Trifluoroacetoxy-2-methylpentane	3.115	71.05
2	Propanoic acid, 2-hydroxy-2-methyl-	3.929	59.05
3	Cyclopropyl methyl carbinol	4.168	58.05
4	Carbamic acid, 2-(dimethylamino)ethyl ester	4.354	58.05
5	Butanoic acid, 2-hydroxy-, ethyl ester	4.427	59.05
6	2-Methylpentyl formate	4.455	71.05
7	1-Heptanol	4.565	70.10
8	Octane, 3,3-dimethyl-	4.656	71.10
9	3-Heptanol, 4-methyl-	4.780	59.05
10	Propane, 1,3-dichloro-	4.817	76.00
11	Dodecane, 2,6,10-trimethyl-	5.156	57.05
12	Dodecane, 4,6-dimethyl-	5.236	57.10
13	Ethane, hexachloro-	5.392	116.90
14	Dodecane, 2,6,10-trimethyl-	5.800	57.05
15	Naphthalene	6.993	128.10
16	Tetradecane	7.265	57.05
17	(Z),(Z)-2,4-Hexadiene	7.332	77.00
18	Decane, 2-methyl-	7.399	57.05
19	Benzaldehyde, 2,4-dimethyl-	7.455	133.10
20	Undecane, 4,8-dimethyl-	7.510	71.10
21	Tridecane	7.817	57.05
22	Tridecane	7.893	57.05
23	Benzene, 1,3-bis(1,1-dimethylethyl)-	7.983	175.15
24	Nonadecane	8.092	57.05
25	Dodecane, 4,6-dimethyl-	8.316	71.10
26	Nonadecane	8.431	57.05
27	Dodecane, 4,6-dimethyl-	8.507	71.10
28	Dodecane, 2,6,10-trimethyl-	8.615	71.10
29	Dodecane, 4,6-dimethyl-	8.940	71.10
30	Naphthalene, decahydro-1,4a-dimethyl-7-(1-methylethyl)-, [1S-(1.alpha.,4a.alpha.,7.alpha.,8a	9.200	57.05
31	Benzene, 1-cyclobuten-1-yl-	9.841	129.10
32	Hexadecane	9.903	57.05
33	Dodecanal	10.039	57.05
34	Heptadecane	10.184	57.05
35	Cyclotetrasiloxane, octamethyl-	10.471	281.05
36	Hexadecane	10.570	57.05
37	Dodecane, 2,6,10-trimethyl-	10.687	71.10
38	Heptane, 2,4-dimethyl-	10.720	85.10
39	Hexadecane, 1-bromo-	10.852	57.05
40	Undecane, 2,4-dimethyl-	11.005	85.10
41	Octane, 2-methyl-	11.039	71.10
42	Hexadecane	11.110	71.10
43	Tetradecane	11.157	57.05
44	Octadecane, 1-iodo-	11.220	57.05
45	2,4-Di-tert-butylphenol	11.349	191.15
46	Hexadecane	11.653	57.10
47	Dodecanoic acid	11.918	73.05
48	Octadecane	12.008	57.10
49	Hexadecane	12.346	57.10
50	1,4-Methanoazulen-9-ol, decahydro-1,5,5,8a-tetramethyl-, [1R-(1.alpha.,3a.beta.,4.alpha.,8a.	12.425	85.10
51	Heptadecane	12.515	57.05
52	Pentadecane, 4-methyl-	12.590	71.10
53	Diphenylamine	12.653	169.10
54	Heptadecane	12.749	57.05
55	Hexadecane, 2,6,10,14-tetramethyl-	13.057	57.05
56	Heneicosane	13.192	57.05
57	Heneicosane	13.249	57.05
58	Hexadecane	13.330	57.05
59	Dodecane, 1-iodo-	13.385	57.05
60	Heneicosane	13.483	57.05
61	Hexadecane	13.540	71.10
62	Heneicosane	13.588	71.10
63	Hexadecane	13.945	57.05
64	Heneicosane	14.058	71.10
65	Pentacosane	14.156	57.05
66	3,5-di-tert-Butyl-4-hydroxybenzaldehyde	14.250	219.15
67	Octadecane, 1-iodo-	14.320	57.05
68	Heneicosane	14.546	57.05
69	Octacosane	14.645	57.10
70	6-Octen-1-ol, 3,7-dimethyl-, acetate	14.966	68.05
71	Heneicosane	15.042	57.05
72	1-Tetradecanamine	15.151	59.05
73	Phytol	15.213	57.05
74	1,2-Benzenedicarboxylic acid, bis(2-methylpropyl) ester	15.302	149.05
75	Phytol	15.397	57.05
76	Hexadecane	15.445	57.05
77	Hexadecane	15.489	57.05
78	Tetracosane	15.650	57.10
79	Dotriacontane	15.695	71.10
80	Tetracosane	15.738	267.05
81	7,9-Di-tert-butyl-1-oxaspiro(4,5)deca-6,9-diene-2,8-dione	15.835	57.05
82	Benzoic acid, 2-benzoyl-, methyl ester	16.006	163.05
83	n-Hexadecanoic acid	16.202	73.05
84	Dibutyl phthalate	16.441	149.05
85	Heneicosane	16.544	57.05
86	Palmitic acid, TMS derivative	17.010	117.10
87	Heneicosane	17.115	57.10
88	Dotriacontane	17.246	57.05
89	Heneicosane	17.476	57.05
90	Tetrapentacontane	17.564	57.05
91	Phytol	17.631	71.10
92	Tetracosane	17.725	57.05
93	Tetrapentacontane	17.817	71.10
94	Octadecanoic acid	18.056	57.05
95	Tetrapentacontane	18.116	71.10
96	Tetrapentacontane	18.190	71.10
97	Docosane	18.371	57.05
98	4-Morpholinepropanamine	18.495	100.05
99	3-Isopropyl-2,5-piperazine-dione	18.583	114.10
100	Heptadecane, 2-methyl-	18.945	57.05
101	Tetrapentacontane	18.997	57.10
102	Heneicosane	19.229	57.05
103	Tetracosane	19.450	57.05
104	Tetrapentacontane	19.529	71.10

## Data Availability

The data used to support the findings of this study can be made available by the corresponding author upon request.

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
