# Peer review of "Phytochemical Constituents, Antimicrobial Properties and Bioactivity of Marine Red Seaweed (Kappaphycus alvarezii) and Seagrass (Cymodocea serrulata)"

_foods, 2023, doi:10.3390/foods12142811_

Round 1
Reviewer 1 Report
This study evaluated the chemical composition of two important marine plants with antioxidant and antimicrobial activities. My comment as follow:
1. Inadequate background explanation in the ‘introduction’ section. What are the shortcomings of the current studies on the phytochemical composition as well as the activity of these two marine plants? Or in other words the importance of this study is not highlighted.
2. The third part of ‘results and discussion’ is perhaps better divided into several parts with different subheadings.
3. In the author's discussion of the results, there is only a list of the reported results in the literature, and there is no further analysis of why they are different from what has been reported or what is the significance of such differences. For example, lines: 238-257.
4. All data in the article is presented in a table, perhaps some can be replaced by images, for example Table 2.
5. The characterization of antioxidant and antimicrobial activity in this study appears to be somewhat monotonous.

It is worth noting that this manuscript needs a careful streamlining of the language.
Author Response
With respect to the comments from the Reviewer 1:
- Inadequate background explanation in the ‘introduction’ section. What are the shortcomings of the current studies on the phytochemical composition as well as the activity of these two marine plants? Or in other words the importance of this study is not highlighted.
According to the suggestion from the Reviewer, it was included a background explanation in the Introduction. Specifically, the following paragraph was included:
“Marine organisms are valuable sources of bioactive compounds for both food and pharmaceutical industries. Bioactive compounds can be obtained from a wide range of marine foods. Nowadays, more than 36,000 compounds with potential interest for human health have been isolated from marine organisms [1]. Significantly, such bioactive compounds can minimize chronic non-communicable disease risk by reducing the onset of inflammation and oxidation [2]. In recent years, seaweeds were reported as an important source of bioactive compounds [3,4]. Another marine organism which contains a large variety of bioactive compound are seagrasses, that although are less used than seaweeds, has been used as a food and as a medicine by coastal region people [5].”
Additionally, in the latest part of the Introduction section, it was included the paragraph:
“Although the composition of K. alvarezii was previously reported even form the same origin and that the phytochemical composition of C. serrulata was also previously reported, it should be considered that both composition of properties of seaweed and seagrasses widely varies depending on a large variety of factors [4,5]. Thus, in a scenario in which important factors such as ocean acidity, salinity or temperature are changing [21], it is important to have updated data on seaweed and seagrasses composition and bioactivities.”
- The third part of ‘results and discussion’ is perhaps better divided into several parts with different subheadings.
Thank you very much for your comment. In the revised version of the manuscript, it was included subheadings in the Results and discussion section, including “Phytochemical analysis”; “antioxidant activity”; “antimicrobial activity” and “presence of bioactive compounds”
- In the author's discussion of the results, there is only a list of the reported results in the literature, and there is no further analysis of why they are different from what has been reported or what is the significance of such differences. For example, lines: 238- 257.
Thank you very much for your comment. In the revised version of the manuscript, it was included information about effects of the phytochemical groups discussed. In this sense, it was included the following paragraphs:
“Both phenolic compounds and tannins have important roles as bioactive compounds. Phenols have important antioxidant, antimicrobial, antiinflamatory and anticancer activ-ities [29, 30], whereas tannins were reported to have antiviral, antibacterial and antioxi-dant activity [31,32].”
“With respect to glycosides, it was reported that they have antioxidants and antiinflammatory activities which find application in prevention and managements of several human diseases [31]. Flavonoids are also an important a phtochemical group due to its antimicrobial, antiviral, antioxidant and spasmolytic activities [32]”.
Consequently, it was also included the following references in the refences list to reinforce the information provided:
- Rengasamy, R.R.K.; Arumugam, R.; Perumal, A. Seagrasess as potential source of medicinal food ingredients: Nutritional analysis and multivariate approach. Biomed. Prev. Nutr. 2013, 3(4), 375-380.
- Kytidou, K.; Artola, M.; Overkleeft, H.S.; Aerts, J.M.F.G. Plant glycosides and glycosidades: A treasure-trove for therapeutics. Front. Plant. Sci. 2020, 11, 357.
- All data in the article is presented in a table, perhaps some can be replaced by images, for example Table 2.
Thank you very much for your comment. We think that in the information placed in Table 2, the difference between mm of inhibition is small and the differences may not be well observed. Instead, in order to illustrate the article, we have included as Figure 1 a photograph of the two species investigated.
- The characterization of antioxidant and antimicrobial activity in this study appears to be somewhat monotonous.
Thank you very much for your comment. In fact, there were not made an expanded investigation of the antimicrobial properties, including a long variety of microorganism or including several methods of inhibition detection. In the same way, antioxidant properties are described by basic methods. However, we believe that it should be taken into account that the objective of the investigation of these parameters is not to carry out an extensive research, but to demonstrate, on the basis of the phytochemical composition, that these properties really exist. Obviously, a more extensive research on both parameters could be carried out and we believe that it would be interesting, but they should be investigated in more specific articles on those aspects, and not in one that includes such a wide range of parameters as this one.
Author Response
With respect to the comments from the Reviewer 2:
The article "Phytochemical constituents, antimicrobial properties and bio- 2 activity of marine red seaweed (Kappaphycus alvarezii) and 3 seagrass (Cymodocea serrulata)" by Deep Das, Abimannan Arulkumar , Sadayan Paramasivama Aroa López-Santamarina and Alicia del Carmen Mondragón is devoted to the actual problem of the composition and properties of algae.
We wish to thank the reviewers for his revision of the manuscript. We expect our answers to be the adequate to take the paper into consideration and that it can be published in Foods.
There are many inaccuracies and inconsistencies in the work.
Thank you for your comments. The manuscript has been revised to correct the inconsistencies.
The paper is written carelessly.
Thank you for your comments. The manuscript has been thoroughly revised, and its wording has been carefully corrected. In fact, in the original version it were some unproper capital letters and some references were not correctly formatted in the main text.
In Abstract, Latin names should be written in italics.
Thank you for your comments. That is correct, there were Latin words that were misspelled, these words have been corrected.
95- what were the transportation conditions of the samples?
Thank you for your question. The samples were transported to the laboratory in a portable cooler protected from sunlight within 1 h after collection. This information has been added to the new manuscript. Page 3, line 98-99.
97- where were the samples dried? 106- where were the samples stored at this temperature?
Thank you for your questions. The samples have been dried in an oven. This information has been added to the new manuscript (Page 3, line 101). On the other hand, the samples were stored at room temperature to avoid enzymatic reaction due to high temperatures or moisture uptake due to low temperatures.
170- what type of water bath was used? 174- on what scales were the samples weighed?
Thank you for your comments. A thermostatic bath with stirring was used. An analytical balance was used to weigh the sample. This information has been added in the new manuscript (Page 4, line 178).
176- 3000°C-4500°C for 3-5 h which oven model can tolerate these temperatures, the temperatures are very high
Thank you for your comments. Obviously, it was a mistake. In the revised version of the manuscript temperatures range were changed to “500-550 ºC”
Table 1.- 1.63±2.73 why is the error more than the result? 22.1±01 not clear error 278- 1.88 ± 011 mg/g error is not clear
Thank you for your comments. The errors have been corrected in the new manuscript. In the first case, “2.73” was changed to “0.73”, that is the correct value. In the other cases, a dot was omitted in each case in the original version of the manuscript and in the revised form it was corrected to “22.1±0.1” and “1.88 ± 0.11” mg/g, respectively.
339- dodecanal, dodecanal, how do these compounds differ
Thank you for your comments. It was a mistake; the compound was repeated. It has been corrected in the new manuscript.
Table 3- Retention Time what is the unit of this value
Thank you for your comments. The units of retention time are minutes. This information has been added to the new manuscript.
Reviewer 3 Report
Overall, the abstract provides a clear overview of the study's objectives and methods. However, there are a few suggestions for improvement:
- Provide a context: It would be helpful to provide a brief background or context for why the evaluation of marine organisms' phytochemical constituents, antioxidant activity, and antibacterial properties is important. This would help readers understand the significance of the research.
- Structure the introduction: The introduction can be organized into subsections to improve readability. Consider dividing it into separate sections to cover topics such as the importance of seaweeds and seagrasses, their nutritional composition, bioactive compounds, and their potential health benefits. This will make it easier for readers to navigate the information.
- Clarify the research gap: Highlight the existing knowledge gap or research need that this study aims to address. This will help readers understand the novelty and relevance of the research.
- Expand the keywords: Include additional relevant keywords that reflect the key aspects of the study, such as "phytochemical analysis," "antioxidant activity," "antibacterial properties," and "nutritional value."
- Check grammar and formatting: Ensure consistent punctuation, capitalization, and formatting throughout the abstract and introduction to enhance readability and professionalism.
- Overall, the abstract provides a clear overview of the study's objectives and methods. However, there are a few suggestions for improvement:
- Provide a context: It would be helpful to provide a brief background or context for why the evaluation of marine organisms' phytochemical constituents, antioxidant activity, and antibacterial properties is important. This would help readers understand the significance of the research.
- Structure the introduction: The introduction can be organized into subsections to improve readability. Consider dividing it into separate sections to cover topics such as the importance of seaweeds and seagrasses, their nutritional composition, bioactive compounds, and their potential health benefits. This will make it easier for readers to navigate the information.
- Clarify the research gap: Highlight the existing knowledge gap or research need that this study aims to address. This will help readers understand the novelty and relevance of the research.
- Expand the keywords: Include additional relevant keywords that reflect the key aspects of the study, such as "phytochemical analysis," "antioxidant activity," "antibacterial properties," and "nutritional value."
- Check grammar and formatting: Ensure consistent punctuation, capitalization, and formatting throughout the abstract and introduction to enhance readability and professionalism.
Overall, the manuscript provides a good foundation for investigating the phytochemical constituents, antimicrobial properties, and bioactivity of marine red seaweed and seagrass. However, it is important to complete the manuscript by including the results section, discussion, and conclusion. Additionally, addressing the comments and suggestions mentioned above will improve the clarity and overall quality of the manuscript.
minor issues
Author Response
With Respect to the comments from the Reviewer 3:
Overall, the abstract provides a clear overview of the study's objectives and methods. However, there are a few suggestions for improvement:
- Provide a context: It would be helpful to provide a brief background or context for why the evaluation of marine organisms' phytochemical constituents, antioxidant activity, and antibacterial properties is important. This would help readers understand the significance of the research.
Thank you for your comments. In the revised version of the manuscript, it was included as a brief background the following paragraph: “Marine organisms are valuable sources of bioactive compounds for both food and pharmaceutical industries. Bioactive compounds can be obtained from a wide range of marine foods. Nowadays, more than 36,000 compounds with potential interest for human health have been isolated from marine organisms [1]. Significantly, such bioactive compounds can minimize chronic non-communicable disease risk by reducing the onset of inflammation and oxidation [2]. In recent years, seaweeds were reported as an important source of bioactive compounds [3,4]. Another marine organism which contains a large variety of bioactive compound are seagrasses, that although are less used than seaweeds, has been used as a food and as a medicine by coastal region people [5].”
- Structure the introduction: The introduction can be organized into subsections to improve readability. Consider dividing it into separate sections to cover topics such as the importance of seaweeds and seagrasses, their nutritional composition, bioactive compounds, and their potential health benefits. This will make it easier for readers to navigate the information.
Thank you for your comments. In the revised version of the manuscript, the Introduction section was reorganized according to the reviewer´s suggestion.
- Clarify the research gap: Highlight the existing knowledge gap or research need that this study aims to address. This will help readers understand the novelty and relevance of the research.
According to the suggestion from the Reviewer, it was included a background explanation in the Introduction. Specifically, the following paragraph was included:
“Marine organisms are valuable sources of bioactive compounds for both food and pharmaceutical industries. Bioactive compounds can be obtained from a wide range of marine foods. Nowadays, more than 36,000 compounds with potential interest for human health have been isolated from marine organisms [1]. Significantly, such bioactive compounds can minimize chronic non-communicable disease risk by reducing the onset of inflammation and oxidation [2]. In recent years, seaweeds were reported as an important source of bioactive compounds [3,4]. Another marine organism which contains a large variety of bioactive compound are seagrasses, that although are less used than seaweeds, has been used as a food and as a medicine by coastal region people [5].”
Additionally, in the latest part of the Introduction section, it was included the paragraph:
“Although the composition of K. alvarezii was previously reported even form the same origin and that the phytochemical composition of C. serrulata was also previously reported, it should be considered that both composition of properties of seaweed and seagrasses widely varies depending on a large variety of factors [4,5]. Thus, in a scenario in which important factors such as ocean acidity, salinity or temperature are changing [21], it is important to have updated data on seaweed and seagrasses composition and bioactivities.”
- Expand the keywords: Include additional relevant keywords that reflect the key aspects of the study, such as "phytochemical analysis," "antioxidant activity," "antibacterial properties," and "nutritional value."
All the keywords suggested by the Reviewer were included in the revised version of the manuscript. However, please note that Food´s instructions for authors claims that the number of keywords should be 3-10. In order to include the keywords suggested by the reviewer, in the revised version of the manuscript it were deleted the keywords “antimicrobial”; “phenolic” and “GC-MS”
- Check grammar and formatting: Ensure consistent punctuation, capitalization, and formatting throughout the abstract and introduction to enhance readability and professionalism.
Thank you for your comments. Formatting aspects as capital letters, punctuation, etc, were checked and corrected in the revised version of the manuscript.
- Overall, the abstract provides a clear overview of the study's objectives and methods. However, there are a few suggestions for improvement: Provide a context: It would be helpful to provide a brief background or context for why the evaluation of marine organisms' phytochemical constituents, antioxidant activity, and antibacterial properties is important. This would help readers understand the significance of the research.
Although the composition of K. alvarezii was previously reported even form the same origin and that the phytochemical composition of C. serrulata was also previously reported, it should be considered that both composition of properties of seaweed and seagrasses widely varies depending on a large variety of factors [4,5]. Thus, in a scenario in which important factors such as ocean acidity, salinity or temperature are changing [21], it is important to have updated data on seaweed and seagrasses composition and bioactivities
- Structure the introduction: The introduction can be organized into subsections to improve readability. Consider dividing it into separate sections to cover topics such as the importance of seaweeds and seagrasses, their nutritional composition, bioactive compounds, and their potential health benefits. This will make it easier for readers to navigate the information.
Thank you for your comments. In the revised version of the manuscript, the Introduction section was reorganized according to the reviewer´s suggestion.
- Clarify the research gap: Highlight the existing knowledge gap or research need that this study aims to address. This will help readers understand the novelty and relevance of the research.
According to the suggestion from the Reviewer, it was included a background explanation in the Introduction. Specifically, the following paragraph was included:
“Marine organisms are valuable sources of bioactive compounds for both food and pharmaceutical industries. Bioactive compounds can be obtained from a wide range of marine foods. Nowadays, more than 36,000 compounds with potential interest for human health have been isolated from marine organisms [1]. Significantly, such bioactive compounds can minimize chronic non-communicable disease risk by reducing the onset of inflammation and oxidation [2]. In recent years, seaweeds were reported as an important source of bioactive compounds [3,4]. Another marine organism which contains a large variety of bioactive compound are seagrasses, that although are less used than seaweeds, has been used as a food and as a medicine by coastal region people [5].”
Additionally, in the latest part of the Introduction section, it was included the paragraph:
“Although the composition of K. alvarezii was previously reported even form the same origin and that the phytochemical composition of C. serrulata was also previously reported, it should be considered that both composition of properties of seaweed and seagrasses widely varies depending on a large variety of factors [4,5]. Thus, in a scenario in which important factors such as ocean acidity, salinity or temperature are changing [21], it is important to have updated data on seaweed and seagrasses composition and bioactivities.”
- With respect to the comments about “Expand the keywords: Include additional relevant keywords that reflect the key aspects of the study, such as "phytochemical analysis," "antioxidant activity," "antibacterial properties," and "nutritional value."”
All the keywords suggested by the Reviewer were included in the revised version of the manuscript. However, please note that Food´s instructions for authors claims that the number of keywords should be 3-10. In order to include the keywords suggested by the reviewer, in the revised version of the manuscript it were deleted the keywords “antimicrobial”; “phenolic” and “GC-MS”
- With respect to the comments about “Check grammar and formatting: Ensure consistent punctuation, capitalization, and formatting throughout the abstract and introduction to enhance readability and professionalism.”
Thank you for your comments. Formatting aspects as capital letters, punctuation, etc, were checked and corrected in the revised version of the manuscript.
Overall, the manuscript provides a good foundation for investigating the phytochemical constituents, antimicrobial properties, and bioactivity of marine red seaweed and seagrass. However, it is important to complete the manuscript by including the results section, discussion, and conclusion. Additionally, addressing the comments and suggestions mentioned above will improve the clarity and overall quality of the manuscript.
The authors really appreciate the positive comments from the reviewer. All of the above comments have been incorporated into the manuscript, including aspects of the discussion. We believe that thanks to the positive comments from the reviewers, the manuscript has gained in quality during the review process.
Round 2
Reviewer 2 Report
The article "Phytochemical constituents, antimicrobial properties and bio- 2 activity of marine red seaweed (Kappaphycus alvarezii) and 3 seagrass (Cymodocea serrulata)" by Deep Das, Abimannan Arulkumar , Sadayan Paramasivama Aroa López-Santamarina and Alicia del Carmen Mondragón
I received full responses to my comments. The authors have taken into account the reviewers' comments, but there are still some mistakes:
Line 335 - 1.88 ± 011 mg/g – error correction.
The authors did not specify in all places in the paper the models of the equipment on which the experiments were performed.
Table 4 – NO – correct NO

Author Response
Thank you very much for your valuable comments. Please find a detailed response to your comments:
The article "Phytochemical constituents, antimicrobial properties and bio- 2 activity of marine red seaweed (Kappaphycus alvarezii) and 3 seagrass (Cymodocea serrulata)" by Deep Das, Abimannan Arulkumar, Sadayan Paramasivam Aroa López-Santamarina and Alicia del Carmen Mondragón I received full responses to my comments. The authors have taken into account thereviewers' comments, but there are still some mistakes:
Line 335 - 1.88 ± 011 mg/g – error correction.
Thank you for your comment, it was a mistake and was changed in the revised version of the manuscript to “1.88 ± 0.11 mg/g”
The authors did not specify in all places in the paper the models of the equipment on which the experiments were performed.
According to the suggestion from the reviewer, models and manufacturer of all equipment, even small equipment such as scales or water baths, were included. The think that now all relevant equipment were included in the revised version of the manuscript
Table 4 – NO – correct NO
Thank you for your comment. In the revised version of the manuscript, “NO” was changed to “No.”